# Turning Points as a Catalyst for Escaping Partner Violence: A Shelter-Based Phenomenological Study Examining South African Women’s Experiences of Leaving Abusive Relationships

**DOI:** 10.3390/ijerph22060880

**Published:** 2025-05-31

**Authors:** Annah Mabunda, Mathildah Mpata Mokgatle, Sphiwe Madiba

**Affiliations:** 1Department of Public Health, School of Health Care Sciences, Sefako Makgatho Health Sciences University, Pretoria 0208, South Africa or mabundaannah01@gmail.com (A.M.); or mathildahmokgatle1@gmail.com (M.M.M.); 2Independent Researcher, 1 Theart Street, Orchards, Pretoria 0182, South Africa

**Keywords:** abuse, intimate partner violence, leaving process, turning points, triggers, shelters, South Africa

## Abstract

Even though South Africa is a signatory to international declarations that aim to eliminate violence against women, intimate partner violence (IPV) remains a significant problem. While many women eventually leave IPV relationships after enduring violence for a long period, the matter of why women leave IPV relationships is not adequately researched in South Africa. This study explored the turning points that trigger the decision to leave IPV relationships and examined the process of leaving in a sample of women who left their abusive partners. In-depth interviews with 11 women living in shelters of safety for women in Gauteng Province, South Africa, were analyzed using Giorgi’s phenomenological analysis method. Leaving IPV relationships was a complex process that involved multiple decisions and actions over time; thus, most women endured many years of abuse. Leaving was triggered by various turning points, resulting in the leaving process being either planned or unplanned. For some of the women, the turning point was a specific violent event, while, for others, it was a culmination of violent events. Severe or escalating violence and the desire to protect their children from the impact of violence were key turning points for most of the women, such that they feared for their lives and that of their children. Overall, turning points played a crucial role in the decisions to leave abusive partners, and the leaving process for women was characterized to a great extent by fear. Understanding the complexities of the process of leaving and the relevance of turning points is essential to informing the development of appropriate interventions to respond more appropriately to women experiencing IPV.

## 1. Introduction

Although most sub-Saharan African (SSA) countries have agreed to many international declarations that aim to eliminate violence against women, gender-based violence (GBV), including intimate partner violence (IPV), remains a significant health challenge. IPV is one of the most common forms of violation experienced by women globally [1]. The World Health Organization (WHO) defines IPV as acts of physical aggression, sexual coercion, psychological abuse, and controlling behaviors within an intimate relationship that cause physical, sexual, or psychological harm [2]. The most common forms of IPV include physical, emotional, and sexual violence.

IPV is widespread throughout much of SSA with an overall prevalence of 36%, exceeding the global average of 30% [3]. A recent systematic review and meta-analysis reported an overall estimate of 44.4% for the prevalence of IPV among women aged 15–49 years of age [4]. The prevalence of all types of IPV is consistently higher in SSA compared to other global regions [5], and global estimates show that more women in Africa are subject to lifetime IPV (45.6%) than women anywhere else in the world [3].

The prevalence of IPV in South Africa ranges from 20% to 50% in studies in which women report having experienced IPV at some point in their lives [6,7]. The 2016 South African Demographic and Health Survey (SADHS) indicates that approximately 20% of women have experienced physical violence, while 17% have reported emotional violence and 6% sexual violence. The National Department of Health estimates that 26% of ever-partnered South African women have experienced some form of IPV at some point in their lives [8]. The high prevalence of IPV among women in SSA is sustained by strong and rigid harmful gender norms such as the tolerance of partner abuse and acceptance of wife-beating [9].

IPV has both direct and indirect lifelong impacts on the health of women in abusive relationships. There are direct pathways, such as injuries, and indirect pathways, such as chronic health problems that arise from prolonged stress and long-term mental health consequences [10,11]. Since IPV can vary in frequency and severity and occurs on a continuum, ranging from one episode to chronic and severe episodes over the years [12], its implications are far-reaching. IPV affects the physical and overall well-being of women, women’s productivity, and ability to care for themselves and their families [13], infant mortality and morbidity, and the social fabric of society in a significant way [14,15].

Although the literature shows that IPV is progressive [16,17] and the effects are cumulative over a long time [18], women often stay longer in IPV relationships [19,20] due to the many barriers that they face to leaving IPV relationships. Research provides different explanations for the reasons why women leave IPV relationships. The decision to leave IPV may be influenced by factors such as having a safe place to go to, available external support, and child-related and personal factors [21,22,23]. However, women’s decision making around leaving IPV may be extremely challenging [24]. Moreover, the leaving process is complex and shaped by individual, familial, economic, and sociocultural factors [22,23,25,26].

Research indicates that making changes to leave IPV is a gradual process involving multiple stages [27,28]. IPV-related healthcare research has increasingly used the Transtheoretical Model of Change (TTM) to highlight how women make decisions to leave IPV [24,27,29]. Hence, there is consensus among researchers that the stages described in the TTM are applicable to IPV [27,30]. The TTM asserts that women move through a series of stages in the leaving process. Within IPV research, stages of change (SOC) are foundational constructs of the TTM consisting of five stages (precontemplation, contemplation, preparation, action, and maintenance). These stages are identified as a pathway that women navigate as they work toward achieving safety [24].

In addition, researchers have reported on turning points or triggers that facilitate women’s decision to leave abusive partners [27,31,32]. The turning point concept explains specific incidents, factors, or circumstances that trigger or prompt women to begin the process of leaving IPV relationships. These are changes occurring in the lives of women experiencing IPV at a specific time which permanently change how they view the violence and their relationship, which ultimately leads them to leave the abusive partner [32]. Chang and colleagues describe turning points as dramatic shifts in beliefs and perceptions of themselves, their partners, and/or their situation that alter the women’s willingness to tolerate the situation and motivate them to consider change [27]. Women may reach a turning point when they experience extreme or escalating violence, fear of being killed, a desire to protect children from violence, increased awareness of the IPV, and the perception of the violence as unbearable [33,34,35].

Although researchers have reported on the many barriers that women face in leaving IPV relationships [19,20,36,37], many women eventually leave their abusive partners [38]. However, the matter of why women leave IPV relationships and the process of leaving are not adequately researched particularly in SSA countries such as South Africa [38], even though the country is a signatory to international declarations that aim to eliminate violence against women. South Africa has implemented policies, strategies, and community-based interventions to mitigate IPV. However, the country remains one of the highest-ranking countries, with an IPV prevalence of 20% to 50%. The lack of empirical data has implications for the success of interventions to mitigate IPV by providing tailored support to women during their journey to leave their abusive partners. The need to support and empower women in making informed decisions to leave IPV is essential to a comprehensive response that addresses their needs [21].

There is a dearth of studies that investigate what triggers the decision to leave an abusive partner in South Africa and other SSA countries. Research conducted in developed countries highlights the importance of turning points in the process of leaving IPV relationships [28,31,32]. However, studies conducted in developed countries may not fully account for South African women’s decision to leave IPV relationships given the different personal, social, and cultural contexts of IPV [16,20,37].

Therefore, this study aimed to explore the turning points that trigger the decision to leave IPV relationships and examined the process of leaving in a sample of women who left their abusive partners. The purpose of the study is to gain a deeper understanding of the leaving process from the point of contemplating leaving until the final break up from an abusive partner.

## 2. Materials and Methods

### 2.1. Study Design and Setting

The study employed a descriptive qualitative phenomenological approach using in-depth interviews. As a qualitative research approach, phenomenology emphasizes understanding the essence of human experiences as described by individuals who have lived through the phenomenon in question. Descriptive phenomenology is best suited to exploring the turning points and the process of leaving IPV relationships as it focuses on describing the essence of a lived experience [39,40].

The study took place in four shelters for abused women located in Gauteng Province, South Africa. These shelters provide short-term interventions for women and children in crisis, offering safe spaces and a range of supportive services, including counseling, basic needs, and various forms of skill development. Within the South African context, shelters operate under the Department of Social Development’s Victim Empowerment Programme. The shelters employ full-time social workers to provide ongoing support. Women gain access to these shelters through self-referral, referrals from South African Police Services (SAPS), or referrals from community members. While the duration of the stay in shelters ranges from 3 to 12 months, extensions are possible in cases where women lack safe accommodation after the stipulated duration of the stay period. Staying in the shelters is free for unemployed women; however, a minimal fee is charged for those who are employed.

### 2.2. Sample

We used purposive sampling to select women who have experienced the phenomenon of interest [41]. In this study, women who had experienced IPV and had left their violent partners were selected with the help of the resident social workers and house mothers in the shelters. To ensure that the sample reflects a diverse group of participants, we employed maximum-variation sampling. We included women with short- and long-duration IPV relationships and different lengths of stay in the shelters to provide a range of perspectives.

### 2.3. Data Collection

Data were collected from 11 women using in-depth interviews guided by a semi-structured interview guide prepared in English and translated into IsiZulu and Sepedi to accommodate their preferred languages. The lead author, also referred to as the interviewer, conducted all the interviews. The interviewer asked broad questions to explore the process of leaving violent partners and the turning points that influenced leaving. In line with phenomenological inquiry [40], the researcher asked follow-up questions and used probes to further explore the phenomenon to understand the women’s experiences of leaving IPV relationships.

All interviews were conducted in the shelters in a private room in the preferred language of the participants to allow them to speak freely about their experience of IPV, the leaving process, and turning points. Each interview was audio recorded with consent from the participants and lasted for about 60–90 min. Written informed consent was obtained before the interviews after the researcher thoroughly explained the voluntary nature of participation, the right to withdraw from the study without penalties, confidentiality, and the use of pseudonyms.

In addition to the interviews, a brief biographical questionnaire captured the demographic information of the participants such as age, level of education, employment status, marital status, IPV duration, race, number of children, income sources, place of residence, and relevant biographical details about their partners.

### 2.4. Data Analysis

Data were analyzed inductively using Giorgi’s phenomenological analysis method [42] by the authors who met regularly to reflect on the emerging analysis. The lead author translated verbatim transcripts into English and prepared the transcripts for data analysis. Giorgi’s phenomenological analysis method aims to uncover the meaning of a phenomenon as experienced by an individual through the identification of essential themes. The method involves five key steps. The first step involves the authors familiarizing themselves with the data by listening to the recordings and reading transcripts numerous times. Next, the authors identify themes that describe the process of leaving abusive partners to develop an overall sense of the data. During the third step, the authors grouped meanings into clusters of themes that were common across participants’ experiences. In step four, the transformation of meaning units into descriptive expressions was performed. A second level of coding then generated sub-themes and connections between these themes. The NVivo 12, a qualitative data analysis package, was utilized for analysis. Finally, these themes were integrated into exhaustive descriptions of the phenomena.

To ensure rigor in data collection and analysis, we employed several strategies, including using a good audio recorder, conducting peer-debriefing sessions, prolonged engagement, and keeping an audit trail [43]. The authors also met regularly during data analysis and interpretation to reflect on the emerging themes and analysis to reduce investigator bias [44]. Reflexivity is an integral part of qualitative research: it contributes to transparency and rigor, enhancing the credibility of study findings. Reflexivity was used to check on the integrity of the data by acknowledging the authors’ own values, preconceptions, assumptions, and potential sources of bias that may influence the research process. The lead investigator has substantial experience working with women in IPV. As the key interviewer, she kept a reflective journal throughout data collection process to reflect on how her prior experiences with IPV, as well as her personal beliefs, motivations, and assumptions, may influence data collection and the study participants. She used bracketing to avoid bias by remaining true to the facts given by the interviewee during data collection. Overall, reflexivity ensured the credibility of the findings, with the authors continuously reflecting on how their perspectives may influence the interpretation of findings, thereby reducing researcher bias and their influence on data analysis and interpretation [39,45].

By acknowledging and actively engaging with their subjectivity, researchers can minimize the impact of personal perspectives on the research process, reducing the risk of biased interpretations and conclusions.

### 2.5. Ethical Considerations

Ethical approval for the study was received from the Research Ethics Committee of Sefako Makgatho Health Sciences University (SMUREC/H/309/2020: PG). The Department of Social Development and the Managers of the selected shelters provided permission to conduct the study. The participants signed informed consent forms before the interviews. The women’s names were anonymized, and no personal identifiers were collected to further protect the women’s privacy. Arrangements were made with the resident social workers at the shelter to provide counseling if necessary due to concern that participants might be distressed while reliving their experiences during the interview.

## 3. Results

### 3.1. Demographic Profiles of Participants

Eleven women aged between 20 and 48 years were interviewed. Seven reported their relationship status as single, three as married, and one as engaged. Four reported to have completed secondary education, three had post-secondary qualifications, and two did not complete secondary education. The duration of the relationships ranged from six months to 15 years, with an average length of stay of 5.7 years. The length of stay in shelters ranged from three weeks to six months (Table 1).

Table 2 presents details of the study sample during their stay in relationships characterized by IPV. Eight women reported their marital status as single; they were in cohabitation relationships with their abusive partners. Only three women were married. Seven women reported that they were unemployed, three were formerly employed, and two were self-employed. The source of income for those who were unemployed was either child support grants or financial support from their partners. All but one woman had children either from an earlier relationship or with the man in question; one woman did not have children.

The partners’ profiles described by the women varied in terms of age, educational attainment, employment status, and housing ownership. Their age ranged from 22 years to 60 years with an average age of 35 years. The partners’ educational attainment was higher than that of the women; for instance, one individual held a law degree and ventured into property development, while another was a chartered accountant. Some of the men had stable jobs, whereas others relied on income from self-employment activities. Their financial status was better than that of the women, most of whom were unemployed. Thus, before the women moved to the shelters, seven stayed with their partners in their partners’ rented homes or in the backyard of the partner’s parental home. Those who were married lived in rented flats where the couple contributed to the rental payment. Only one woman stayed with her unemployed partner in a place that she rented.

### 3.2. Themes

Three themes and eight sub-themes explaining the process of leaving a violent relationship and the turning points that influenced the decision to leave emerge from the data analysis (Table 3).

#### 3.2.1. Deciding to Leave Violent Relationships

The data in this study revealed that women hesitated to leave abusive relationships due to uncertainty about whether it was the right decision and because of fears about their future. As a result, the decision became highly emotional for them.


*I was always in two minds thinking, should I or should I not go, is this a good decision, where am I going to go. It is a very emotional decision (Maria).*



*I was scared and uncertain about what was going to happen next. Am I making the right decision? Should I stay and maybe work it out or maybe leave and see what you can do in the future? It was a very emotional decision (Maria).*


Planned move

The narratives from some of the women revealed that leaving was often a well-planned move that also included preparations such as discussing the plan to leave with significant others, searching for an alternative place to stay, and buying necessities for future use. Emotional detachment was also reported as they were preparing to leave.


*I thought that this was not the kind of life I wanted to live. I am going to leave, wherever I go I will see, I cannot stay and live this kind of life am living. I remember telling one of my friends before I left the house that I wanted a place to stay but I didn’t have money to rent, I remember going to this shack they said it was ZAR 4000 to ZAR 4500 (Tshidi).*



*I bought extra furniture. I bought a bed and kitchen unit. I started buying second-hand furniture, as I was preparing myself to leave him (Keamogetse).*



*I was thinking most of the time that should this behavior continue, I would run away from this house even if I didn’t know where I’d find refuge. If it comes to that, I will sleep by the police station. It was something I was plotting bit by bit. I wanted to escape without a trace. I realized that I could not be in this relationship. I let him walk out thinking he would find me when he returned. That is when I escaped; I took my things and ran for my life (Masello).*


Others detached themselves emotionally before physically moving as they plan to escape.


*I think it is the right decision that I left him because I am tired of him. I left him a long time ago while staying with him (Tebogo).*


Unplanned move

The narratives of the women revealed that some of them abruptly left the relationships as the violence intensified out of fear for their safety. They left their belongings behind, leaving without knowing where they were going.


*I decided I was going to leave now, or I was going to die. Remembering the actual abuse, I knew that if he got into the flat, I promise I wouldn’t have been here today (Maria).*



*I then said to myself, today I am leaving, I don’t know where I am going. I want to run away… I am leaving no matter what. I’d rather sleep under the bridge; I think it is the best thing to do because I will have peace, but I am not going to stay here any longer (Dikeledi).*


Women made attempts to leave the violent relationship; however, they ended up returning to their abusive partners many times before they finally decided to end the relationship. They contemplated leaving more than once; they left and came back to the place they shared with their partner until they eventually left the violent relationship. Their narratives show that their coming back was influenced by their partner apologizing, having no permanent place to stay, and not being welcomed when they returned to their parents’ home.


*I’ve been living with him, and after two years, I left, but though he did me so bad, I would always go back to him and hope that it’s going to be fine. When I got back, things were worse actually (Dipuo).*



*Level 5 COVID-19 restrictions require a person walking in the streets to have a permit. There were no permits for being chased away. So, I had to go back and ask for accommodation. Getting there, he had no remorse. He called a woman in front of me. He told me straight that it was my fault. He cheats, but it’s my fault. How is it my fault? (Keamogetse).*



*I went back to my partner because my mother was always angry at me. She liked asking me when I was going back to my partner, I felt like my time was over at home. She was always complaining about everything, especially about money (Mapule*
*).*


A few women were compelled to leave their relationships not by choice, but because their abusive partners chased them away. They left without having had time to plan because their partners ordered them to leave and never return.


*He chased me away, he once strangled me and said certain things like I want to chop you and put you in a suitcase. He chased me away, and I left without shoes, but I had slippers (Mapule).*



*I left because he was using all the money to buy drugs and not food. I left for two days and came back. When I arrived, he asked me where I was, I told him I had gone to see my friends. He then chased me away and said I must never come back. I refused to go, and he started beating me. I then left forever (Puleng).*


#### 3.2.2. Turning Points

The turning point concept explains circumstances that triggered or prompted the women to begin the process of leaving IPV relationships. Three distinct concepts encompass the concept of turning points: awareness of the violence, escalation of violence, fear of harm to self and of children, and fatigue and loss of hope

Escalation of violence

Women reported that the escalation of the violent acts, the intensity, and frequency, were essential promoters of the decision to leave the violent partner. Some women in the study felt that the violence had become intolerable, and they decided to end the relationship.


*The decision to leave came when the violent acts and situation became worse. That is `when I made up my mind that I am leaving him, I am leaving (Dikeledi)*



*The third time when it happened that is what made me decide, that was the worst beating. It was only the slaps at first and secondly, I saw that he was not willing to change, it meant I had to make a choice now, either stay in the relationship and be beaten like nothing, I had to make a choice and live like that or die in this relationship, I had to move away, I knew that I can live without him that is what made me make a choice, the beating was more than the other beatings (Tshidi).*



*Things became very, very bad because he strangled me, he was like, it’s either you die or no man’s going be with you. That’s when I left him, he almost strangled me to death, but I was like, I am, I am not staying here in this stage (Dipuo).*


Awareness of the violence

Women stayed with their abusive partners and were not aware that what was happening to them was abuse. Realizing that their relationship was abusive added to the decision to leave their partners.


*I realized that the life that we were living was not normal. This is not love. I realized that it had nothing to do with that love but something to do with me. When he comes back from work he will swear at me for no reason, he will be complaining about everything. There are things I cannot change. I asked myself, “What exactly am I doing here?” “What is it am I hoping to find in this guy?” “Does he have that? He proved many times that he didn’t want me (Keamogetse).*



*I’m glad I left because he was beating me in public and humiliating me in front of people. He kept on beating me in the street, sometimes I would run into the streets with the upper part of my body uncovered (Puleng*
*).*


Fear of harm and death

Fear of harm or death was a turning point for the participants, who realized that the relationship was ruined after confronting death. Women reported having experienced severe violent acts and threats to their life several times. This made them scared and fear for their lives. The threat to their life sharpened, and they decided to escape from the violent relationship.


*The reason I left him was because he had promised to kill me. I realised that I was going to die now, it was clear that he would end up killing me and my child or else kill me and leave my child suffering while I was gone (Lebogang)*



*When I decided to leave, it was the second time that he started breaking the windows and threatening to break the gates, I decided I was going to leave now, or I was going to die. (Maria).*



*I realized that he would eventually kill me, or I would be poisoned by the crystal. He was beating me in front of the child and promised to kill me (Puleng).*


Fear for the lives of the children

Many women in the study are mothers, and protecting their children was a high priority for them; this meant some of them had to leave. The risk of their children enduring abuse or suffering after their demise haunted them.


*I left because there was a risk that my child could end up being abused (Lebogang).*



*I made a decision and said, “If I die, who is going to look after my child’? I decided to stop him, and I left (Tebogo)*


Fatigue and loss of hope

Part of the reason for women in the study to leave violent relationships was the recognition that the partner was not going to change their violent behaviors. Growing intolerance to abuse and a consequent weariness resulting from their continuous experience of violence by their partner also contributed to women leaving violent relationships.


*So, things were just worse. Cause every time I fight for myself, remember I am in his yard, the aunties will come the father and everybody. I was like, I’m leaving everything, including my son. I was like, it’s fine I’ve fought enough (Dipuo).*



*Since I came back to him again, I never found happiness. I started to feel again that I was losing myself. I was fighting that I must not lose my son and must not lose myself. By the time I was doing this, I thought it was enough, and I was tired, and I said I couldn’t live this kind of life. I thought it would happen in a good way (leaving) that I would get a job and find myself a place to stay (Mapule).*



*I thought he would change, he loves me. I felt sorry for him. Then I thought what’s the use of feeling sorry for him because in the end, I’m the one who gets hurt (Tebogo)*



*We were always fighting. I had enough about him now. I don’t love him anymore I don’t want anything to do with him (Puleng).*


#### 3.2.3. Support from Significant Others

Several participants reported seeking the intervention of family, friends, neighbors, police services, and other community members. The women received financial assistance to travel back home or to shelters, while others were assisted to find shelters. The women mentioned the important role significant others played in their decision to leave their abusive partners. They not only encouraged and supported them to leave but also provided them with practical information, tangible support, and advice that they required.


*My prayer friend shared some contacts with me, she said get out before this gets too far, take a step back (Dipuo).*



*My partner’s older brother would say I should call the Cops. I felt like he knew the support I needed (Keamogetse).*



*I went to my friend’s mother’s and asked for money for the taxi. She gave me the money and said I should pack my bags quickly and she asked her husband to come with her to assist me to pack so my partner would not come back and find me. She called my mother and explained the situation (Tebogo).*


Women who made multiple attempts to leave the violent relationship reported receiving a gradual decrease in support from their families and friends.


*Many people that I told said I should fix my relationship with my partner, others said I should go to report to the police which I did. The police would allow me to sleep at the police station and go home on Monday. Sometimes they would beat him and reprimand him. After some time, he will beat me again and I will go to the police again. Police officers are human beings too, they get tired. I ended up not going to them again (Puleng).*



*I used to go and report to the police and go back to him. Later, whenever I go report, the police would say “It’s a normal case you’ll get back to him” (Lebogang).*



*My friend was like I didn’t think you would even let things get this far when I’ve shared so much with you. Every time I advise you to go here and here and here, you do not. So, I don’t think I’m going to take this as I love you so much that I cannot watch you just destroy your own life because of just a random guy. Until today, our relationship has never been fixed (Dipuo)*


Several of the women’s accounts highlighted that police services were crucial in helping them leave their abusive partners. Once services were accessed, they assisted with safety and assistance to move to shelters.


*I went to the police and related the story, they then made means to serve him with the protection order. They then took me to the shelter with my kids (Keamogetse).*



*I went to the police station and that is when they told me about this place. When I reported the abuse, they asked whether I needed shelter, and I said yes. They told me about a place of safety where I would be staying for a certain period, and they brought me here (Masello).*


## 4. Discussion

The purpose of the study was to explore turning points that trigger the decision to leave to gain a deeper understanding of the processes involved in leaving intimate partner violence. The study focused on women who left a relationship and lived in shelters of safety. Most of the women were unemployed and were single and cohabitating; they thus depended on their partners for financial support and accommodation during their stay in IPV relationships. Overall, most of the women endured all forms of violence for a long period before the final breakup with their abusive partners. The average length of stay in an IPV relationship was 5.7 years. The longer stays in IPV can be explained by the dependence of women on their partners for financial support and accommodation, as most of them were unemployed.

Although they reported experiencing sexual abuse, psychological abuse, and controlling behaviors, physical violence was the most prevalent form of violence. The study found that leaving the IPV relationship was a complex process that involved multiple decisions and actions over time. The process of leaving the IPV relationship has been described as a progressive process rather than as a single event; it is a non-linear process characterized by multiple phases that sometimes overlap [38]. As is consistent with other research findings, we found that leaving a violent relationship was a complex and emotional process marked by a continuum of events [22,23,25,26,31].

The process of leaving described by the women in the current study is consistent with the sequence of stages of the TTM. The TTM posits that women move through a series of stages in the process of deciding to leave an IPV relationship. The stages encompass a continuum that goes from a stage of precontemplation, in which there is no acknowledgment of abuse or contemplation of the ending the relationship; to contemplation, where there is acknowledgment of the abuse and consideration of ending the relationship; to preparation, involving the decision to leave the partner and taking the necessary steps to do so; to action, which is the act of leaving the partner; and, lastly, maintenance, which is the continued disengagement from the abusive partner [24,46].

Our findings indicated that, during the precontemplation stage, the women tried to resolve the violence they experienced. They had no intention of leaving their abusive partners, hence their efforts to adapt their behaviors to become what their partners wanted them to be, to prevent further violence. They also used silence and passiveness, threatening police action, and behaving differently as strategies to avoid the escalation of violence. These strategies were reported in previous research [47,48,49,50]. The women in this study stayed for long periods of time in an IPV relationship without recognizing that they were in an abusive relationship. This suggests that they were in the precontemplation stage for longer periods and were not considering leaving their abusive partners. According to Catallo, there are no set time periods for the individual to move from stage to stage, explaining why some women remain in the precontemplation stage for many years [30]. Therefore, the precontemplation stage is an important phase for creating awareness about IPV.

We found that, while in the contemplating stage, the women began to acknowledge the abuse and made several attempts to leave. They left their abusive partners on multiple occasions only to return after a temporary separation. Their behavior is consistent with the cyclical nature of the TTM process of change, which asserts that individuals may progress and relapse between stages [27]. Previous research reported on the phenomenon of leaving and returning to abusive partners [21,28,33,38].

The study findings revealed that planning was an important phase in the leaving process, during which some of the women planned to leave. The planning that the women adopted is in line with the preparation stage in the TTM, the women made the decision to leave their abusive partners. They planned their exit from the violent relationship overtime and in secret for fear of angering their abusive partners. Their narratives indicated that they emotionally detached themselves from their partners and the relationship long before physically leaving. Bermea and colleagues argue that, during the planning phase, women engage in mental preparation where they emotionally disconnect from their abusive partners [51]. Planning also included discussing the plan to leave with significant others, contacting professionals for help, searching for a safe place to stay, and buying necessities for future use. The study findings are consistent with previous research findings [33,52].

The last stage of the TTM model is action: during this phase, the women actively prepared and executed the plan when they were emotionally ready. As stated above, the decision to leave an abusive relationship is complex [32]; thus, the women endured IPV for long periods before reaching a breaking or turning point that triggered them to begin the process of leaving, which is the preparation stage of the TTM. The women described turning points as the cumulation of violent acts, leading to a point when they knew that they had to take action and leave their abusive partners. They understood that staying within the violent relationship was no longer viable. The women in the current study, as in previous studies, experienced multiple turning points [27,32], not just a single isolated incident that compelled them to leave their abusive partners.

We found that escalation of violence was one of the triggers that the women in this study cited as the turning point. The escalation of violent acts, the intensity and frequency of violent acts, and the fear of being killed compelled them to leave. The feeling that the abuse had become intolerable led them to decide to end the violent relationship. The women felt that the violence had become intolerable when they experienced extreme or escalating violence, such as severe physical abuse, threats of harm, or severe incidents of abuse. This finding is consistent with other research findings [21,28,31,32]. The women started to fear for their lives and believed the death threats from their partners and chose to leave before being killed. Dziewa and colleagues also reported that a moment of extreme fear might trigger a break-up with an abusive partner [33].

Awareness and recognizing that their relationship was violent was another turning point. Other studies also reported that recognition of violence as IPV was a turning point for escaping IPV [28,33,53,54]. In keeping with the findings reported by Baholo and colleagues, at the time the women left their abusive partners, they had come to realize that the abuse went against the general characteristics of a loving relationship [21]. They changed their perceptions and started to believe that there was no love in their relationship. They gained insights into IPV through formal support from significant others and professionals [53].

After enduring violence for a long time, the women reached their tipping point due to fatigue and loss of hope that their partner would change. There was a gradual shift in how they perceived their partners as they identified the relationship as abusive. The women stated that they were no longer willing to normalize abuse and felt that they had no love left for the abusive partner. This finding aligns with other research findings reporting that women left IPV when they reached the point where they were tired of dealing with the violence [21,33].

Nearly all the women had children, either from an earlier relationship or with their current abusive partner. The presence of children played a critical role in the women’s decision to leave a violent relationship. The fear of the impact of IPV on their children and the desire to protect their children from harm was a major turning point in leaving their abusive partners. The women feared the risk of their children enduring and witnessing the abuse and that children would suffer in the likelihood that their partners kill them. The thought of their partner killing them and leaving their children orphaned haunted them to the point that they left the violent relationship. As is consistent with previous research [31,55], women left violent relationships because they feared for the lives of their children [28,52,56]. Heron and colleagues found that a crucial turning point for some of the women in their study was a desire to prevent their children from thinking that the abuse was normal [57].

There is a consensus in research that the presence of children represents a double-edged factor, both promoting and hindering the process of leaving violent relationships [31,38]. In the same context, this study found that, while the presence of children in IPV relationships led to longer stays, at a later stage, the presence of children became the reason why women left IPV relationships. Women who stayed longer did so because they were reluctant to separate children from their fathers, whereas others left because they feared for their children’s lives. Similarly, Scheffer et al. reported that the women in their study described how, at first, children were a restraining factor, but they gradually became a driving force to leave the relationship [38].

Our study found that, while the process of leaving the relationship was planned for most of the women, for others, leaving the IPV relationship was not planned but happened abruptly. This was largely because the women realized that the violence was escalating, and they feared for their lives and those of their children. These findings are consistent with previous research showing that women may leave a relationship abruptly at a point of crisis [33,51]. Two women fled urgently and left without knowing where they were going because they were forced out of their homes by their partners. The women in a study conducted by Murray and colleagues [56] described the termination of their relationships by their abusive partners as turning points.

Leaving abruptly has implications for the psychosocial well-being of women. As stated above, during the preparation stage, women engage in mental preparation before finally exiting the IPV relationship. Moreover, those who planned to leave highlighted the importance significant others played in their decision to leave their IPV relationships. They received encouraging words to escape the IPV from family and friends, gained a better perspective on their violent relationship, learned about the availability of shelters, were exposed to external sources of support such as SAPS, and learned about the counseling support available for women in IPV.

The current study and previous research findings [28] show that, while the women benefited from formal and informal sources of help, the leaving process was led by their deliberate decisions. Estraladoh and colleagues suggested that the decision to leave IPV is dependent on women’s perception of the gains and losses inherent in their relationship [31]. Although the decision to leave IPV was based on decisive decisions in response to various turning points that triggered their motivation to leave, the leaving process is aligned to the stages of change model.

The study has a few limitations. First, it had a relatively small sample size of 11 women recruited from shelters in one province in South Africa. Despite the small sample size, the study contributes to a growing knowledge base regarding the factors that may underlie the difficult decision to leave violent relationships and an understanding of women’s perceptions of their turning points within the South African context. Furthermore, despite the small sample, the rich quality of the data obtained from the women provided in-depth insights into the process of leaving abusive partners and contributed to an increasing knowledge base on the complex nature of the decision to leave IPV. Our findings are compatible with research from developing and developed countries; thus, we reported similar factors that can trigger a turning point in a woman’s decision to leave IPV. Nevertheless, future research could benefit from studying diverse geographic locations and larger samples to gain a more detailed understanding of the complex process of leaving abusive partners and factors that influence turning points. Although maximum variation was used and women with different durations in IPV and shelters were sampled, the sample remained homogenous in terms of sexual orientation, race, gender, and geographical location. This limits the generalizability of the findings to other groups such as diverse ethnic, religious, and same-sex couples, as well as men in IPV relationships. Additional research should focus on quantitative approaches to establish differences between different groups in terms of their reasons to leave IPV relationships using larger samples.

## 5. Conclusions

Overall, our findings agree with those for different populations in developed countries, showing that women’s decision to leave their abusive partners is triggered by turning points. We found that, for some women, the turning point was a specific violent event that led to the decision to leave, whereas, for others, there was a culmination of violent events wherein they feared for their own lives and those of their children. Thus, for most women, the leaving process was characterized largely by fear.

To the investigators’ knowledge, this study was the first of its kind in South Africa to explore turning points in IPV. While the turning points in our study are consistent with those of studies conducted in developed countries, the study provides a better understanding of turning points and the process of leaving IPV in a diverse albeit small sample of women using the stages of change model. Insights into turning points are critical for healthcare professionals to tailor interventions and counseling to respond more appropriately to women experiencing IPV. Interventions should take into consideration the social and cultural contexts of IPV.

Understanding the complexities of the process of leaving IPV relationships is essential to informing the development of targeted and appropriate interventions that address the needs of women who experience IPV. It is envisaged that this deeper understanding of and knowledge about the leaving process for women experiencing IPV will assist healthcare professionals as well as other key stakeholders such as social workers and nongovernment organizations working with women to develop sensitive responses to the needs of women.

## Figures and Tables

**Table 1 ijerph-22-00880-t001:** Participants’ demographic profiles.

Age	Participants	Length in Relationship	Participants	Length in Shelter	Participants	Level of Education	Participants
20 yrs	3	6 months	1	1 month	4	Tertiary	3
24 and 25 yrs	2	2 yrs	2	2 months	2	Student	1
27 yrs	1	3 yrs	1	4 months	1	Grade 12	4
30 yrs	1	4 yrs	2	5 months	1	Secondary	3
38 and 39 yrs	2	5 yrs	2	6 months	3		
44 yrs	1	8 yrs	1				
48 yrs	1	14 and 15 yrs	2				

**Table 2 ijerph-22-00880-t002:** Details of study sample during their stay in an IPV relationship.

Marital Status	Participants	Employment Status	Participants	Source of Income	Participants	Number of Children	Participants
Single	8	Employed	3	Salary	5	None	1
Married	3	Self employed	2	Child support grant	2	1	7
		Unemployed	6	Partner support	1	3	2
				Partner support//child support grant	3	4	1

**Table 3 ijerph-22-00880-t003:** Summary of themes and subthemes.

Themes	Subthemes
Deciding to leave violent relationships	Planned move
	Unplanned move
Turning points	Escalation of the violence
	Awareness of the violence
	Fear of harm and death
	Fear for the lives of the children
	Fatigue and loss of hope
Support from significant others	

## Data Availability

The data that support the findings of this study are available from the corresponding author, upon reasonable request.

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
