# Peer review of "Turning Points as a Catalyst for Escaping Partner Violence: A Shelter-Based Phenomenological Study Examining South African Women’s Experiences of Leaving Abusive Relationships"

_ijerph, 2025, doi:10.3390/ijerph22060880_

Round 1
Reviewer 1 Report
Comments and Suggestions for Authors
The article is compelling in that it offers an in-depth exploration of women's decisions to leave abusive relationships. The concept of turning points is particularly valuable for understanding these decisions. However, certain aspects related to environmental influences may have been overlooked—specifically, the cultural and contextual factors that shape women's experiences in these situations. In my view, the relevance of the study and the significance of its findings should be more clearly justified.
Regarding the study sample, it is relatively small and limited, which raises concerns about the generalizability of the results to a broader population of women in similar circumstances. More detailed information should be provided about the sociodemographic characteristics of the 11 women interviewed, particularly in terms of their cultural and environmental contexts. This would help clarify the extent to which these external factors—beyond support from significant others—may have influenced their decisions.
Moreover, while the article concludes that women’s decisions to leave abusive partners are diverse, complex, and shaped by multiple factors, such a broad conclusion is not fully supported by the limited sample. These findings, while insightful, cannot be generalized without caution.
Accordingly, I recommend that the article be revised in the following ways:
- Provide a stronger justification for the study’s relevance and clarify the specific contributions it makes to the field.
- Include more comprehensive information about the sample and relevant contextual data.
- Revise the conclusions to accurately reflect the scope and limitations of the study’s findings.
Author Response
We thank the reviewer for the comments and suggestions. We have implemented all the comments to the best of our abilities
Comments
The article is compelling in that it offers an in-depth exploration of women's decisions to leave abusive relationships. The concept of turning points is particularly valuable for understanding these decisions. However, certain aspects related to environmental influences may have been overlooked—specifically, the cultural and contextual factors that shape women's experiences in these situations. In my view, the relevance of the study and the significance of its findings should be more clearly justified.
Regarding the study sample, it is relatively small and limited, which raises concerns about the generalizability of the results to a broader population of women in similar circumstances. More detailed information should be provided about the sociodemographic characteristics of the 11 women interviewed, particularly in terms of their cultural and environmental contexts. This would help clarify the extent to which these external factors—beyond support from significant others—may have influenced their decisions.
Moreover, while the article concludes that women’s decisions to leave abusive partners are diverse, complex, and shaped by multiple factors, such a broad conclusion is not fully supported by the limited sample. These findings, while insightful, cannot be generalized without caution.
Accordingly, I recommend that the article be revised in the following ways:
- Provide a stronger justification for the study’s relevance and clarify the specific contributions it makes to the field.
We have revised the justification of the study to the best of our ability.
- Include more comprehensive information about the sample and relevant contextual data.
We added a table (Table 2) to provide additional information on the women during their stay in IPV. We also provide data on the partners’ profiles as described by the women.
- Revise the conclusions to accurately reflect the scope and limitations of the study’s findings.
We revised the conclusion to address the objectives of the study and reflect the study findings. In the limitations, we acknowledged that the study sample was small but also highlighted that the similarities of the study findings with those of studies conducted in developed countries underscore the factors that trigger turning points.
Reviewer 2 Report
Comments and Suggestions for Authors
The current manuscript explores IPV relationship turning points in a sample of South African women. As the authors state, IPV is a global phenomenon and the exited of those relationships could be better understood to facilitate tailored intervention approaches. This qualitative study helps do this. However, I have some questions about the coding themes that I think, if answered, would improve the conceptualization of the turning points process.
-
Minor edit needed on line 68 in the manuscript. I think “women often stay long in IPV relationships” should read “women often stay longer…”.
-
The last sentence that starts on line 71 is incomplete.
-
Line 91 is redundant.
-
What was the average length of a IPV relationship for participants?
-
Authors might want to consider including a reflexivity statement in the manuscript given that they engaged with the process during coding.
-
The subthemes of forced out of the home and unplanned home are presented as distinct, but I am not convinced these need to be separate. Conceptually they read as the same theme.
-
When several quotes are presented together, it is difficult to determine where one quote ends and another begins. Perhaps spacing between quotes or table might help.
-
Is fear of harm and death not also an escalation of violence?
-
The discussion could benefit from connecting the findings to a theoretical rationale for some of the complexities articulated about the termination process.
Author Response
We thank the reviewer for the comments and suggestions. We have implemented all the comments to the best of our abilities
Comments
The current manuscript explores IPV relationship turning points in a sample of South African women. As the authors state, IPV is a global phenomenon and the exited of those relationships could be better understood to facilitate tailored intervention approaches. This qualitative study helps do this. However, I have some questions about the coding themes that I think, if answered, would improve the conceptualization of the turning points process.
- Minor edit needed on line 68 in the manuscript. I think “women often stay long in IPV relationships” should read “women often stay longer…”.
Thanks for pointing this out, we replaced long with longer.
- The last sentence that starts on line 71 is incomplete.
We rephrased the sentece
- Line 91 is redundant.
We deleted the sentence.
- What was the average length of an IPV relationship for participants?
The average length of stay of 5.7 years has been added on page 5, line 220.
- Authors might want to consider including a reflexivity statement in the manuscript given that they engaged with the process during coding.
We expanded on the reflexivity statement, page 5, lines 191 to 201
- The subthemes of forced out of the home and unplanned move are presented as distinct, but I am not convinced these need to be separate. Conceptually they read as the same theme.
We merged the quotes from the subtheme forced out of the home with the quotes under unplanned move.
- When several quotes are presented together, it is difficult to determine where one quote ends and another begins. Perhaps spacing between quotes or table might help.
We agree, however, we followed the journal format, and this is how they want the quotes presented, nevertheless, we added spacing before and after quotes.
- Is fear of harm and death not also an escalation of violence?
We would like to keep them separate because fear is a significant turning point in women in IPV in our study and other studies globally.
- The discussion could benefit from connecting the findings to a theoretical rationale for some of the complexities articulated about the termination process.
We used the Transtheoretical Model of Change (TTM) Model or Stages of Change (SOC) Model to explain the continuum of events that characterise the process of leaving an abusive partner. The SOC model explains behaviour in this case leaving IPV as occurring as a progression through series of stages, and has been used to assess an individual’s readiness to adopt a new healthier behaviour, see page 2, line 76 to 84 and from page 11, lines 444-----
Reviewer 3 Report
Comments and Suggestions for Authors
Overall, the paper was quite well written, with wonderful use of the women's voices, and flowed well.
The only criticism is that the paper sets up that research on the issue is lacking in South Africa, it is not made clear how the women's experiences do differ in this region of the world. Is the conclusion that the turning points are similar to those reported in other areas? Clarity on this and what (if anything) new can be learned from this study would be beneficial.
Author Response
We thank the reviewer for the comments and suggestions. We have implemented all the comments to the best of our abilities.
Comments
Overall, the paper was quite well written, with wonderful use of the women's voices, and flowed well.
- The only criticism is that the paper sets up that research on the issue is lacking in South Africa, it is not made clear how the women's experiences do differ in this region of the world.
Throughout the discussion, we indicated that the study findings are similar or corroborate those of other researchers in developed and developing countries despite the differences in the social and cultural contexts of IPV in South Africa. In the introduction, we highlighted the limited data in South Africa and sub-Saharan Africa on this topic (turning points and leaving process); therefore, most sources used to support our findings are from studies conducted in developed and developing countries.
- Is the conclusion that the turning points are similar to those reported in other areas? Clarity on this and what (if anything) new can be learned from this study would be beneficial.
We added a statement to indicate that the turning points identified in the study are similar to those reported in other settings in developed and developing countries. We further added a statement to indicate the contribution of the study.
Round 2
Reviewer 1 Report
Comments and Suggestions for Authors
The work has improved in the aspects I requested. Therefore, I have accepted it for publication.
Author Response
We appreciate the reviewer for the positive comments.
Reviewer 2 Report
Comments and Suggestions for Authors
The authors have addressed reviewer comments adequately.
Author Response
We thank the reviewer for the positive comments about the status of our manuscript